# Preoperative risk assessment and spirometry is a cost-effective strategy to reduce post-operative complications and mortality in Mexico

Yolanda Mares-Gutiérrez[1,2], Guillermo Salinas-Escudero[3], Belkis Aracena-Genao[4], Adrián Martínez-González[5], Manuel García-Minjares[6], Yvonne N. Flores[7,8,9]*

1 Pulmonary Physiology Department, Hospital General de México Dr. Eduardo Liceaga, Mexico City, Mexico, 2 Universidad de la Salud, Mexico City, Mexico, 3 Centro de Estudios Económicos y Sociales en Salud, Hospital Infantil de México Federico Gómez, Mexico City, Mexico, 4 Instituto Nacional de Salud Pública, Cuernavaca, México, 5 Facultad de Medicina, Departamento de Salud Pública, Universidad Nacional Autónoma de México, Mexico City, Mexico, 6 Coordinación de Universidad Abierta, Innovación Educativa y Educación a Distancia, CUAIEED, Universidad Nacional Autónoma de México, Mexico City, Mexico, 7 Unidad de Investigación Epidemiológica y en Servicios de Salud, Morelos, Instituto Mexicano del Seguro Social, Cuernavaca, México, 8 UCLA Center for Cancer Prevention and Control Research and UCLA-Kaiser Permanente Center for Health Equity, Fielding School of Public Health and Jonsson Comprehensive Cancer Center, Los Angeles, United States of America, 9 UCLA Department of Health Policy and Management, Fielding School of Public Health, Los Angeles, United States of America

* ynflores@ucla.edu

## Abstract

### Aim

Combining preoperative spirometry with the Assess Respiratory Risk in Surgical Patients in Catalunia (ARISCAT) risk scale can reduce post-operative complications and improve patient survival. This study aimed to assess the cost-effectiveness of performing spirometry or not in conjunction with the ARISCAT scale, to reduce post-operative complications and improve survival among adult patients undergoing elective surgery in Mexico.

### Methods

A cost-effectiveness analysis (CEA) was performed to compare the specific cost and health outcomes associated with the combined use of the ARISCAT scale and preoperative spirometry (Group 1), and the use of the ARISCAT scale without preoperative spirometry (Group 2). The health outcomes evaluated were post-operative complications and survival. The perspective was from the health care provider (Hospital General de México) and direct medical costs were reported in 2019 US dollars. A decision tree with a time horizon of eight months was used for each health outcome and ARISCAT risk level.

### Results

The combined use of the ARISCAT scale and spirometry is more cost-effective for reducing post-operative complications in the low and moderate-risk levels and is cost-saving in the high-risk level, than use of the ARISCAT scale without spirometry. To improve patient

**Data Availability Statement:** All relevant data are within the manuscript and its Supporting information files.

**Funding:** The author(s) received no specific funding for this work.

**Competing interests:** The authors have declared that no competing interests exist.

survival, ARISCAT and spirometry are also more cost-effective at the moderate risk level, and cost-saving for high-risk patients, than using the ARISCAT scale alone.

## Conclusions

The use of preoperative spirometry among patients with a high ARISCAT risk level was cost-saving, reduced post-operative complications, and improved survival. Our findings indicate an urgent need to implement spirometry as part of preoperative care in Mexico, which is already the standard of care in other countries.

## Introduction

An estimated 230 million major surgeries are performed globally each year [1]. In Mexico, the number of surgeries performed from 2000 to 2015 increased to 3 million procedures per year [2]. The Instituto Mexicano del Seguro Social (IMSS) performs 50% of these procedures, and they performed almost 1.5 million surgeries between 2018 and 2019 [3,4]. However, in recent years, the SARS-CoV-2 pandemic has had a major impact on the distribution of healthcare resources, with the number of surgeries performed at IMSS dropping to less than one million procedures in 2020 [2]. Of these surgical procedures, 40% resulted in post-operative complications [5–7]. These post-operative complications have been associated with delayed recovery, increase patient morbidity and mortality, longer hospital stays [1,7–13], increased cost of care [14,15] and decreased function or loss of independence [16]. Studies have shown that an adequate preoperative assessment [8] can prevent post-operative complications by identifying and minimizing risk factors, stabilizing comorbidities, and monitoring non-modifiable factors [17].

Currently, a more multidisciplinary approach is being used to identify risk factors in patients who will undergo surgery [18]. The Assess Respiratory Risk in Surgical Patients in Catalonia (ARISCAT) scale is a tool for evaluating preoperative risk based on seven factors: age, pulse oximetry, respiratory infection in the 30 days prior to surgery, anemia, surgical site, surgical duration, and surgery type (elective or emergency) [10,11]. This validated scale classifies patients as low, moderate-, or high-risk, and is considered an excellent clinical tool with external validation [5,19,20].

Airflow obstruction is a preexisting condition of particular concern, which is also associated with post-operative complications, and is diagnosed by preoperative spirometry. Spirometry is considered the gold standard for managing respiratory obstruction and is used to predict post-operative complications [20–22]. However, Mexican clinical guidelines only recommend the use of preoperative spirometry for patients with diagnosed lung disease [23]. The combined use of preoperative spirometry and the ARISCAT scale has been shown to reduce post-operative complications and mortality by improving patient outcomes [24–26], but this strategy has not been evaluated in Mexico. This study aimed to assess the cost-effectiveness of preoperative spirometry for preventing post-operative complications and improving survival at each ARISCAT risk level. We examined clinical data from 9,139 patients who had elective surgery and received either ARISCAT with spirometry or ARISCAT alone; and conducted a cost effectiveness analysis to compare the specific cost and health outcomes associated with both groups.

## Materials and methods

### Study design and population

A cost-effectiveness analysis (CEA) was undertaken to evaluate two preoperative assessment strategies: ARISCAT with spirometry versus ARISCAT without spirometry. This evaluation

was conducted from the perspective of the healthcare provider, Hospital General de Mexico (HGM) for its initials in Spanish), with a time horizon of eight months. The health outcomes selected to measure effectiveness were 1) prevented post-operative complications, and 2) survival. The direct medical cost of elective surgeries incurred by the HGM were estimated in 2019 US dollars. Preoperative assessments, surgical procedures, and in-patient care (including care related to post-operative complications) were used to estimate costs. Data were collected from the records of 9,139 patients who received an ARISCAT score from the Department of Pulmonary Physiology prior to elective surgery between 2013 and 2017. Approval for this study was obtained from the Research and Ethics Committee of the HGM (DIR/18/503F/3/030). This study was conducted in accordance with the Declaration of Helsinki and focused on the analysis of secondary data extracted from medical records. Patient informed consent was not required because the HGM ethics committee waived the requirement for informed consent. The HGM authorized YMG to extract the information from the patient charts that was used for this analysis. All data were fully anonymized using a unique code to identify each patient. The analyzed data did not contain any information that could reveal the identity of the patients [27].

The study included patients who were 20 years or older, underwent elective surgery, and either received an ARISCAT assessment with spirometry, or ARISCAT without spirometry. Individuals with no preoperative assessment, a spirometry of inadequate quality, or postponed surgery were excluded from the study. A total of 2,059 patients who met the inclusion criteria were categorized into two groups: patients with ARISCAT and preoperative spirometry (Group 1, n = 1,306) and patients with ARISCAT and no preoperative spirometry (Group 2, n = 753).

The criteria data for post-operative pulmonary complications was defined according to the same criteria used by the PERISCOPE study (atelectasis, bronchospasm, pleural effusion, pneumonia, respiratory failure, pneumothorax, and pulmonary embolism) [10]. Other postoperative complications included, surgical (hypovolemic shock, sepsis, abdominal pain, fistula, bleeding, paralytic ileus, vascular injury, perforation); metabolic (glycemic dysregulation, hepatic or renal failure); cardiovascular (cardiogenic shock and acute myocardial infarction); neurological (acute vascular events); and vascular (deep vein thrombosis). Mortality was recorded if it was directly related to a complication from the surgery that was performed.

## Calculation of costs

Direct medical costs were estimated using the micro-costing technique [28]. Each clinical service was assigned to a cost center in order to quantify the costs of each consultation with a specialist, surgery, lab tests, and imaging. These costs were categorized by input type: 1) medical personnel (surgical, inpatient, and intensive care); 2) supply costs (medication, laboratory reagents, and medical consumables, etc.); and 3) capital costs.

The unit costs of services and medications were obtained from the HGM's cost scales. Staff salaries were based on the Mexican Ministry of Health's medical, paramedical, and allied health professional pay scales, which are published in the Official Gazette of the Federation of Mexico (*Diario Oficial*).

## Cost-effectiveness analysis

A deterministic, mathematical model of the base case results was represented by a decision tree comparing two groups: ARISCAT with spirometry and ARISCAT without spirometry. The model starts with a hypothetical cohort of patients who 1) need surgery, 2) have a low, moderate, or high, ARISCAT score, and 3) either received or did not receive preoperative

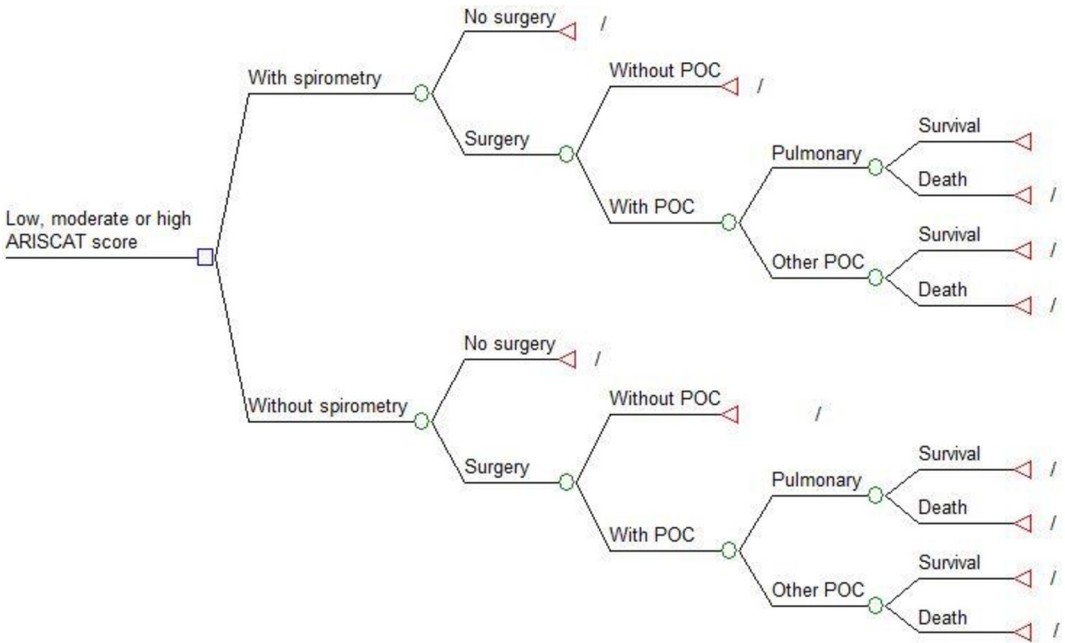

**Fig 1. Decision tree for each ARISCAT risk level with two comparison groups: ARISCAT with and without spirometry.**
HGM costs ($US). *POC, postoperative complications.

spirometry. In the next step of the pathway, the patient's surgery could be cancelled (for reasons unrelated to the spirometry) or proceed. If surgery proceeds, the patient could have no post-operative complications, or develop post-operative complications, and either survive or die. For the model, a total of six decision trees were created—one for each ARISCAT risk level at each health outcome (Fig 1).

A time horizon of eight months was used since the analysis only included the hospitalization time for the assigned surgery. Therefore, application of a discount rate was unnecessary because the period of analysis was less than a year. Willingness-to-pay was based on Mexico's GDP in December 2019 [29]. A GDP between 1 and 3 was cost-effective; below 1, cost-saving; and over 3, not cost-effective [28,30,31].

The average cost-effectiveness ratio (ACER) and the incremental cost-effectiveness ratio (ICER) were calculated for each health outcome at each ARISCAT risk level. The ACER was calculated by dividing cost by effectiveness and the ICER was calculated as (Costs1-Costs2) / (Effectiveness1—Effectiveness2). Post-operative complications in the spirometry group were calculated as follows for the ACER and ICER, respectively:

$$\text{POC ACER without spirometry} = \frac{\text{Cost according to ARISCAT level without spirometry}}{\text{Decrease in POC without spirometry}} \text{ vs}$$

$$\text{POC ICER without spirometry}$$
$$= \frac{\text{Cost according to ARISCAT level without spirometry} - \text{with spirometry}}{\text{Decrease in POC without spirometry} - \text{ with spirometry}}$$

These same formulas were applied to the survival results, for each group with and without spirometry, and for each ARISCAT risk level. These values underwent a univariate sensitivity analysis in which different hypothetical scenarios were posed by increasing or decreasing the

mean, minimum and maximum of each variable by 10%. The triangular distribution was used to estimate the probabilities of the POC and mortality, and costs were estimated using a gamma distribution.

A probabilistic model incorporating Monte Carlo simulations (10,000 runs) was then used to examine the cost-effectiveness plane and acceptability curve distributions. The model's behavior was characterized at two different willingness to pay thresholds (GDP of 1 and 3) [25,32]. The cost-effectiveness analysis was performed using TreeAge Pro v. 2009. The reporting of the study results was developed according to the Consolidated Health Economic Evaluation Reporting Standards (CHEERS) [33].

## Results

Table 1 presents the base case probabilities of post-operative complications (COP) and costs for ARISCAT with spirometry and ARISCAT without spirometry, at each risk level. ARISCAT with spirometry was generally more costly than ARISCAT without spirometry. As the ARISCAT risk level increased, the average cost of care increased.

Table 2 reports the ACER and ICER for post-operative complications and survival scenarios at each risk level. For low-risk patients, the cost of ARISCAT with spirometry ($1,801) was $801 more than the ARISCAT without spirometry ($1,000). Among low-risk patients, ARISCAT with spirometry (76.1%) was 47% more effective than ARISCAT without spirometry

**Table 1. Parameter values of base case model, by ARISCAT risk level, with and without spirometry.** Probabilities and average costs ($US).

| | | ARISCAT Risk Level | | |
| | | Low | Moderate | High |
| Strategy | | Mean (min-max) | Mean (min-max) | Mean (min-max) |
|---|---|---|---|---|
| Probability of Post-operative Complications (POC) | | | | |
| Spirometry | Surgery is cancelled | 0.163 (0.146–0.179) | 0.161 (0.123–0.198) | 0.233 (0.209–0.256) |
| Spirometry | Surgery without POC | 0.915 (0.823–1.006) | 0.867 (0.111–1.622) | 0.871 (0.783–0.958) |
| Spirometry | Surgery with pulmonary POC | 0.54 (0.486–0.594) | 0.132 (0.027–0.236) | 0.791 (0.711–0.870) |
| Spirometry | Surgery survival with pulmonary POC | 0.727 (0.654–0.799) | 0.75 (0.158–1.341) | 0.789 (0.710–0.867) |
| Spirometry | Surgery survival with other POC | 0.535 (0.481–0.588) | 0.571 (0.342–0.799) | 0.4 (0.3–0.4) |
| No Spirometry | Surgery is cancelled | 0.679 (0.611–0.746) | 0.455 (0.123–0.494) | 0.086 (0.077–0.094) |
| No Spirometry | Surgery without POC | 0.910 (0.819–0.999) | 0.815 (0.464–1.165) | 0.43 (0.387–0.473) |
| No Spirometry | Surgery with pulmonary POC | 0.6660 (0.5994–0.699) | 0.603 (0.259–0.946) | 0.569 (0.512–0.625) |
| No Spirometry | Surgery survival with pulmonary POC | 0.95 (0.855–0.999) | 0.75 (0.297–1.203) | 0.604 (0.543–0.664) |
| No Spirometry | Surgery survival with other POC | 0.50 (0.45–0.599) | 0.571 (0.242–0.899) | 0.576 (0.518–0.633) |
| HGM Costs US$ | | | | |
| Spirometry | Probability surgery is cancelled | 4.73 (4.26–5.20) | 4.73 (4.26–5.20) | 4.73 (4.26–5.20) |
| Spirometry | Surgery without POC | $2,133 ($1,912-$2,346) | $2,163 ($1,947-$2,378) | $1,881 ($1,693-$2,069) |
| Spirometry | Surgery survival with pulmonary POC | $2,362 ($2,126-$2,599) | $1,817 ($1,647-$1,999) | $2,691 ($2,423-$2,960) |
| Spirometry | Surgery death with pulmonary POC | $3,603 ($3,243-$3,963) | $1,792 ($1,612-$1,971) | $1,218 ($1,097-$1,340) |
| Spirometry | Surgery survival with other POC | $2,254 ($2,029-$2,479) | $2,975 ($2,677-$3,272) | $2,797 ($2,517–3,077) |
| Spirometry | Surgery death with other POC | $1,477 ($1,329-$1,625) | $1,792 ($1,612-$1,971) | $1,583 ($1,425-$1,742) |
| No Spirometry | Probability surgery is cancelled | $3 ($3-$4) | $3 ($3-$4) | $3 ($3-$4) |
| No Spirometry | Surgery without POC | $2,919 ($2,627-$3,211) | $3,107 ($2,796-$3,418) | $4,214 ($3,793-$4,635) |
| No Spirometry | Surgery survival with pulmonary POC | $2,664 ($2,398-$2,930) | $5,398 ($4,857-$5,937) | $4,661 ($4,195-$5,127) |
| No Spirometry | Surgery death with pulmonary POC | $2,664 ($2,397-$2,930) | $5,624 ($5,061-$6,186) | $4,812 ($4,331-$5,293) |
| No Spirometry | Surgery survival with other POC | $9,644 ($8,679-$10,608) | $3,755 ($3,379-$4,130) | $5,191 ($4,673-$5,710) |
| No Spirometry | Surgery death with other POC | $9,175 ($8,258-$10,093) | $1,592 ($1,433-$1,751) | $3,369 ($3,033-$3,706) |

**Table 2. Cost-effectiveness analysis, by ARISCAT risk level, with and without spirometry.** 2019 HGM costs ($US).

| Strategy | Cost | Incremental Cost | Effectiveness | Incremental Effectiveness | ACER US$ | ICER US$ |
|---|---|---|---|---|---|---|
| **HGM COSTS, US$** | | | | | | |
| **LOW RISK, HGM COSTS** | | | | | | |
| **POC** | | | | | | |
| No spirometry | $ 1,000 | | 29.1% | | $ 3,436 | |
| Spirometry | $ 1,801 | $ 801 | 76.1% | 47% | $ 2,366 | $ 1,704 |
| **SURVIVAL** | | | | | | |
| No spirometry | $ 1,000 | | 99.4% | | $ 1,012 | |
| Spirometry | $ 1,801 | $ 801 | 97.2% | -2.1% | $ 1,839 | -$ 36,655 |
| **MODERATE RISK, HGM COSTS** | | | | | | |
| **POC** | | | | | | |
| Spirometry | $ 1,840 | | 72.7% | | $ 2,528 | |
| No spirometry | $ 1,900 | $ 60 | 44.4% | 28.3% | $ 4,112 | $ 44 |
| **SURVIVAL** | | | | | | |
| Spirometry | $ 1,840 | | 96.3% | | $ 1,921 | |
| No spirometry | $ 1,900 | $ 60 | 97.1% | 0.8% | $ 1,947 | $ 5,077 |
| **HIGH RISK, HGM COSTS** | | | | | | |
| **POC** | | | | | | |
| Spirometry | $ 1,486 | | 66.8% | | $ 2,219 | |
| No spirometry | $ 3,987 | $ 2,501 | 39.3% | -27.5% | $ 10,130 | -$ 9,079 |
| **SURVIVAL** | | | | | | |
| Spirometry | $ 1,486 | | 97.1% | | $ 1,510 | |
| No spirometry | $ 3,987 | $ 2,501 | 75.3% | -21.8% | $ 5,297 | -$ 11,568 |

(29%) to prevent post-operative complications. The ACER for ARISCAT with spirometry was $2,366, compared to $3,436 without spirometry. The ICER (the cost of preventing a post-operative complication) for ARISCAT with spirometry was $1,704, controlling for all other factors, which indicates it is a cost-effective alternative. In terms of improving survival in the low-risk group, ARISCAT with spirometry (99%) was less effective than without spirometry (97%), for a difference of -2%. The cost per life saved with spirometry was $1,839, versus $1,012 without spirometry. This suggests that spirometry is not a cost-effective option to improve survival int the low-risk group, and is therefore dominated by the alternative without spirometry.

In the high-risk group, the ARISCAT with spirometry had a lower mean cost ($1,486) compared to the option without spirometry ($3,987), for a difference of $2,501. For preventing pre-operative complications, ARISCAT with spirometry (66.8%) was more effective than without spirometry (39%), for a difference of -28%. The ACER for ARISCAT with spirometry was $2,219, compared to $10,130 without spirometry. Controlling for all other factors, the ICER for ARISCAT with spirometry was -$9,079, and therefore is a cost-saving strategy.

For improving survival outcomes among high-risk patients, the ARISCAT with spirometry strategy (97%) was more effective than the option without spirometry (75%), for a difference of -22%. The cost per life saved was $1,510 for the ARISCAT with spirometry and $5,297 without spirometry. The ICER for the ARISCAT with spirometry strategy was -$11,568, making it a cost-saving strategy alternative.

Fig 2 shows the sensitivity analysis of the survival model for patients with a moderate ARISCAT risk level. The variables with the most favorable impact on the ICER were: 1) decreasing by 10% the cost of surgery with no preoperative spirometry and no post-operative complications and 2) increasing by 10% the cost of surgery with spirometry and no post-operative

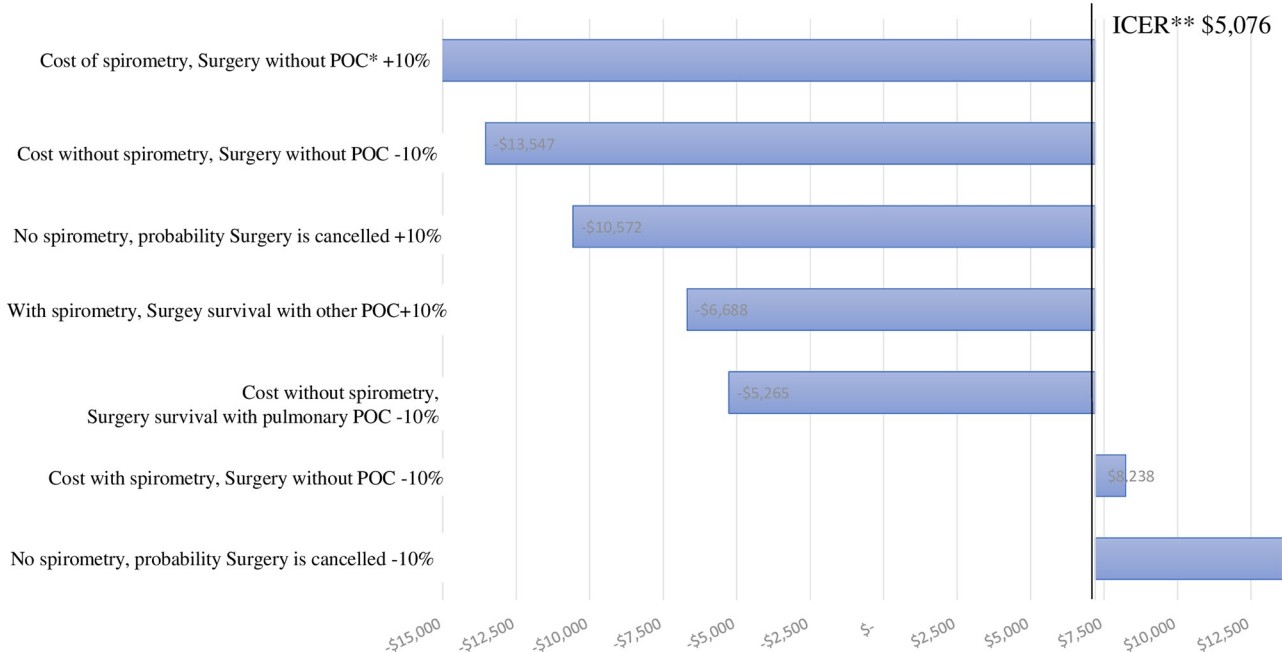

**Fig 2. Sensitivity analysis of survival at moderate risk level.** *POC, postoperative complications, ** ICER, incremental cost-effectiveness ratio.

complications. The results were not robust since the direction of the model changes from cost-effective to cost saving. The only variable that negatively impacted the ICER was decreasing by 10% the probability that surgery with no preoperative spirometry was cancelled since it influences the results and the alternative with spirometry becomes not cost-effective. Sensitivity analyses for the rest of the models generated very robust results and did not affect the directionality of our findings.

Fig 3, shown below, displays the incremental cost-effectiveness of the ARISCAT with spirometry for preventing post-operative complications. Most probabilities for low- and moderate-risk patients were in the cost-effective quadrant, while those for high-risk patients were in the cost-savings quadrant.

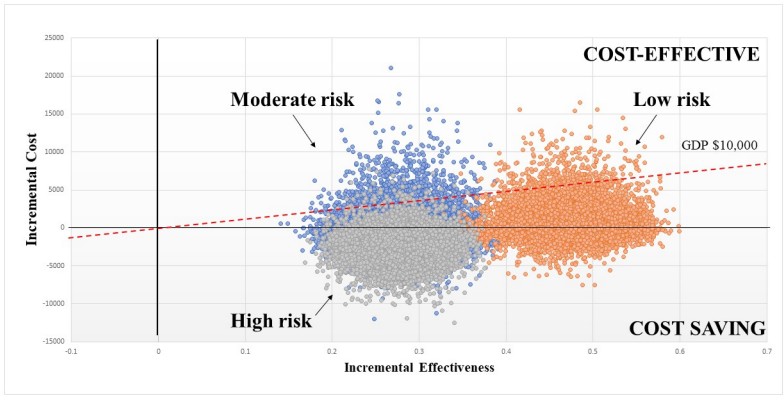

**Fig 3. Cost-effectiveness of spirometry to prevent post-operative complications, by ARISCAT risk level.**

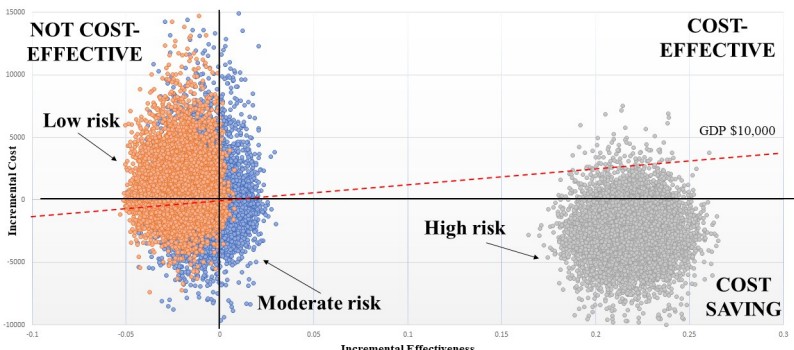

**Fig 4. Cost-effectiveness of spirometry to improve patient survival, by ARISCAT risk level.**

Fig 4, presented below, shows the incremental cost-effectiveness of survival at each ARIS-CAT risk level. While ARISCAT with spirometry is not cost-effective at the low-risk level, it is cost-effective at moderate-risk, and cost-saving at high-risk.

## Discussion

This study assessed the cost-effectiveness of performing spirometry or not, in conjunction with the ARISCAT scale, to reduce post-operative complications and improve survival among adult patients undergoing elective surgery in Mexico. Our results indicate that the combined use of spirometry and the ARISCAT risk scale provides a cost-effective strategy for preventing post-operative complications and death in patients undergoing elective surgery [34–36]. To reduce post-operative complications, ARISCAT with spirometry was found to be a cost-effective strategy in the low- and moderate-risk groups, and cost-saving in the high-risk group. It was also a cost-saving strategy for improving survival in the high-risk group. This suggests that adequately identifying a patient's ARISCAT risk level and conducting a spirometry evaluation on those it will benefit the most could reduce the hospital's costs associated with managing post-operative complications. This aligns with the findings of other research studies that medical costs significantly increase following post-operative complications [11,14,15,37–39].

Among cancer patients, the treatment of post-operative complications that result in death due to conditions such as thromboembolism raised costs by 20%, while the treatment of post-operative respiratory failure ending in death increases costs by over 50% [40]. These findings support the results of our study by demonstrating that the identification of risk factors and comorbidities before surgery lowers the probability of complications and reduces medical costs. Furthermore, performing spirometry on patients with a moderate- and high ARISCAT risk is associated with a decrease in costs, post-operative complications, and a shorter hospital stay.

Although ARISCAT with spirometry was more costly than ARISCAT without spirometry to improve survival, it was more effective among patients with a low or moderate ARISCAT risk and was a cost-saving strategy among high-risk patients. These findings are similar to those reported by Pradarelli et al. [41], which showed that the mortality rate within 30 days of surgery was not impacted by increased costs. This is because a patient's prognosis cannot be improved if their underlying lung disease is not identified and managed, underscoring the importance of diagnosing respiratory disease before surgery.

Additionally, it is important to note that a significant number of patients who had a low ARISCAT risk in our study received a preoperative spirometry evaluation. This practice can overload the department responsible for administering spirometry and limits its availability

for high-risk patients who might undergo surgery without preoperative spirometry, which may put them at increased risk of post-operative complications. Since clinical guidelines in Mexico only recommend the use of preoperative spirometry for patients with diagnosed lung disease, many surgeons conduct incomplete preoperative assessments, which put the health of patients with a high ARISCAT score. Therefore, there is a critical need to improve the quality of preoperative assessment by identifying patients with a moderate or high ARISCAT risk and performing spirometry on higher risk patients, which would optimize the use of available resources, and most importantly, reduce the risk of preventable complications and death.

The main limitation of this study is that it was carried out at a single hospital, limiting the generalizability of the results. However, the sample of patients we examined spanned five-years in a multi-specialty tertiary hospital and the results of the sensitivity analysis were robust, with the direction of cost-effectiveness remaining unchanged in most cases. Additionally, the probabilistic model with the Monte Carlo simulation determined that performing spirometry on patients with a high ARISCAT risk level is cost-saving.

Based on our findings, we suggested the following health policy actions. (1) Train health personnel on preoperative assessment, especially the identification of ARISCAT risk levels. The ARISCAT is quick and simple to administer and helps identify patients who would most benefit from a spirometry evaluation. This clinically beneficial practice improves the quality of care, respects the dignity and rights of patients, and upholds the principles of beneficence and non-maleficence. These cost-saving strategies would also optimize the use of the scarce resources (implicit in all health systems and especially those of low- and middle-income countries) by allowing those resources to be allocated to other health needs. (2) Implement the cost-effective use of preoperative spirometry on patients with a moderate and high ARISCAT risk level, to improve the management of hospital resources, especially those of the Department of Pulmonary Physiology at the HGM. This would allow for preoperative spirometry to be used more efficiently and on patients with the highest need. (3) Recommend the use of preoperatory spirometry for patients with a moderate of high ARISCAT risk in the Mexican Preoperative Clinical Practice Guidelines [22]. Our findings suggest that spirometry should be performed on moderate or high-risk patients, as opposed to the current practice of only patients with previously diagnosed respiratory disease. By identifying more patients who are at-risk of post-operative complications, our policy recommendations would improve equity, efficiency, cost-effectiveness, and the fair distribution and management of resources.

## Conclusions

For preventing post-operative complications, the combined use of ARISCAT with spirometry was cost-effective among low- and moderate-risk patients, and cost-saving in high-risk patients. For improving survival, ARISCAT with spirometry was a cost-saving strategy in the high-risk group. Therefore, we recommend using the ARISCAT scale to assess the risk level of patients undergoing elective surgery and performing preoperative spirometry on high-risk patients as a cost-saving strategy to prevent post-operative complications and increase survival.

The results of this research may help reduce post-operative hospital stays, which could lower the risk of in-hospital infections and associated deaths. In addition, a shorter hospital stay accelerates the process of reintegration into daily life, which contributes to improving the quality of life of patients. The results of this study may also be informative to policy makers who seek to improve the effectiveness, efficiency and quality of surgical services. Finally, we wish to highlight the relevance of our study in terms of its contribution to scientific knowledge, especially since the available evidence on the effectiveness of ARISCAT and spirometry is scarce.

## Supporting information

**S1 Dataset.**
(XLSX)

## Acknowledgments

The authors are grateful to the General Hospital of Mexico, for providing access to the patient medical records and hospital facilities to carry out this research. They also want to thank the Program in Medical Sciences, Dentistry and Health of Master and Doctorate of UNAM and CONACYT, for helping YMG to conduct this research study.

## Author Contributions

**Conceptualization:** Yolanda Mares-Gutiérrez, Adrián Martínez-González, Yvonne N. Flores.

**Data curation:** Yolanda Mares-Gutiérrez.

**Formal analysis:** Yolanda Mares-Gutiérrez, Guillermo Salinas-Escudero, Adrián Martínez-González, Yvonne N. Flores.

**Investigation:** Yolanda Mares-Gutiérrez, Guillermo Salinas-Escudero, Adrián Martínez-González, Yvonne N. Flores.

**Methodology:** Yolanda Mares-Gutiérrez, Guillermo Salinas-Escudero, Adrián Martínez-González, Manuel García-Minjares, Yvonne N. Flores.

**Project administration:** Yolanda Mares-Gutiérrez, Yvonne N. Flores.

**Resources:** Yolanda Mares-Gutiérrez, Guillermo Salinas-Escudero, Adrián Martínez-González, Yvonne N. Flores.

**Software:** Yolanda Mares-Gutiérrez, Guillermo Salinas-Escudero.

**Supervision:** Guillermo Salinas-Escudero, Adrián Martínez-González, Manuel García-Minjares, Yvonne N. Flores.

**Validation:** Yolanda Mares-Gutiérrez.

**Visualization:** Yolanda Mares-Gutiérrez.

**Writing – original draft:** Yolanda Mares-Gutiérrez, Yvonne N. Flores.

**Writing – review & editing:** Yolanda Mares-Gutiérrez, Guillermo Salinas-Escudero, Belkis Aracena-Genao, Adrián Martínez-González, Manuel García-Minjares, Yvonne N. Flores.

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
