## [Decision Letter · Decision Letter 0]

26 Apr 2022

PONE-D-21-36131Preoperative risk assessment and spirometry is a cost-effective strategy to reduce post-operative complications and mortality in MexicoPLOS ONE

Dear Dr. Yvonne N Flores,

Thank you for submitting your manuscript to PLOS ONE. After careful consideration, we feel that it has merit but does not fully meet PLOS ONE’s publication criteria as it currently stands. Therefore, we invite you to submit a revised version of the manuscript that addresses the points raised during the review process.

We look forward to receiving your revised manuscript.

Kind regards,

Carlos Alberto Zúniga-González, Ph.D

Academic Editor

PLOS ONE

Journal Requirements:

2. In ethics statement in the manuscript and in the online submission form, please provide additional information about the patient records/samples used in your retrospective study. Specifically, please ensure that you have discussed whether all data/samples were fully anonymized before you accessed them and/or whether the IRB or ethics committee waived the requirement for informed consent. If patients provided informed written consent to have data/samples from their medical records used in research, please include this information.

The authors appreciate the support they received from the General Hospital of Mexico and the Program in Medical Sciences, Dentistry and Health of Master and Doctorate of UNAM and CONACYT, to conduct this research study. 

Additional Editor Comment:

Dear authors Decision analysis and cost-effectiveness analysis are systematic approaches used to support decision-making under conditions of uncertainty that involve important trade-offs. These mathematical tools can provide patients, physicians, and policy makers with a useful approach to complex medical decision making. Your work is very interesting so I would like you conclusion highlight it as you recoment in line 314.

In line 140 I suggest you specificate the model as mathematical model if it possible.

line 162 specificate the triangular distribution for estimating probabilities

Reviewers' comments:

Reviewer's Responses to Questions

**Comments to the Author**

1. Is the manuscript technically sound, and do the data support the conclusions?

Reviewer #1: Yes

Reviewer #2: Yes

2. Has the statistical analysis been performed appropriately and rigorously? 

Reviewer #1: Yes

Reviewer #2: Yes

3. Have the authors made all data underlying the findings in their manuscript fully available?

Reviewer #1: Yes

Reviewer #2: Yes

4. Is the manuscript presented in an intelligible fashion and written in standard English?

Reviewer #1: Yes

Reviewer #2: Yes

5. Review Comments to the Author

Reviewer #1: 1-The manuscript is technically sound and the data support the conclusions.

2-Consider expanding the description of the statistical techniques used.

3-Consider expanding the data with criteria of complications and mortality.

4-The manuscript is intelligible and is written in standard English

Reviewer #2: The articles contain all the necessary elements for the cost-effectiveness analysis (CEA). It also meets the requirements according to Consolidated Health Economic Evaluation Reporting Standards (CHEERS).

The wording is very coherent and raises the problem from the beginning. The methodology contains the different elements to develop the model and the limitation that it was only carried out in a single hospital is made clear.

6. PLOS authors have the option to publish the peer review history of their article (what does this mean?). If published, this will include your full peer review and any attached files.

Reviewer #1: No

Reviewer #2: No

---

## [Author Response · Author response to Decision Letter 0]

7 Jul 2022

RESPONSE TO PEER-REVIEW REPORT

Name of journal: PLOS ONE

Manuscript ID: PONE-D-21-36131

Title: Preoperative risk assessment and spirometry is a cost-effective strategy to reduce post-operative complications and mortality in Mexico

Corresponding author: Dr. Yvonne N. Flores

We appreciate all the reviewers ‘comments, which have helped to improve our manuscript. The modified lines of text we refer to in our response are from the revised version of our manuscript with track-changes. 

RESPONSE TO SPECIFIC COMMENTS: 

To the Editor: 

1. In ethics statement in the manuscript and in the online submission form, please provide additional information about the patient records/samples used in your retrospective study. Specifically, please ensure that you have discussed whether all data/samples were fully anonymized before you accessed them and/or whether the IRB or ethics committee waived the requirement for informed consent. If patients provided informed written consent to have data/samples from their medical records used in research, please include this information.

Response: As suggested by the Editor, we have added the following information to the Methods section: “Patient informed consent was not required because the HGM ethics committee waived the requirement for informed consent. The HGM authorized YMG to extract the information from the patient charts that was used for this analysis. All data were fully anonymized using a unique code to identify each patient. The analyzed data did not contain any information that could reveal the identity of the patients [28].” Page 8, lines 128-132 (clean file). 

Response: Thank you for the opportunity to clarify this issue. As we mentioned in the Funding Statement section, we received no funding to conduct the research. We have modified the Acknowledgments section to avoid any confusion. The revised version now indicates: “The authors are grateful to the General Hospital of Mexico, for providing access to the patient medical records and hospital facilities to carry out this research. They also want to thank the Program in Medical Sciences, Dentistry and Health of Master and Doctorate of UNAM and CONACYT, for helping YMG to conduct this research study. Page 23, lines 354-357. (clean file) 

We have also removed any funding-related text from the manuscript. 

Response: As requested, we have included our amended statements in our response to the reviewers’ comments.

4. Please note that in order to use the direct billing option the corresponding author must be affiliated with the chosen institute. Please either amend your manuscript to change the affiliation or corresponding author or email us at plosone@plos.org with a request to remove this option.

Response: The corresponding author, Dr. Yvonne N. Flores, (ynflores@ucla.edu) is affiliated with the University of California, Los Angeles, which will cover the publication costs of this manuscript. This information is indicated on the title page of the manuscript, affiliations #7 and #8. Page 2, lines 22-26. (clean file) 

5. Please review your reference list to ensure that it is complete and correct. If you have cited papers that have been retracted, please include the rationale for doing so in the manuscript text or remove these references and replace them with relevant current references. Any changes to the reference list should be mentioned in the rebuttal letter that accompanies your revised manuscript. If you need to cite a retracted article, indicate the article’s retracted status in the References list and also include a citation and full reference for the retraction notice.

Response: As requested, we have reviewed and modified the references so they are complete and correct. We do not cite any retracted papers. 

6. Decision analysis and cost-effectiveness analysis are systematic approaches used to support decision-making under conditions of uncertainty that involve important trade-offs. These mathematical tools can provide patients, physicians, and policy makers with a useful approach to complex medical decision making. Your work is very interesting so I would like you conclusion highlight it as you recommend in line 314.

Response: As requested by the reviewer, we have included the following information to the Conclusions section: “The results of this research may help reduce post-operative hospital stays, which could lower the risk of in-hospital infections and associated deaths. In addition, a shorter hospital stay accelerates the process of reintegration into daily life, which contributes to improving the quality of life of patients. The results of this study may also be informative to policy makers who seek to improve the effectiveness, efficiency and quality of surgical services. Finally, we wish to highlight the relevance of our study in terms of its contribution to scientific knowledge, especially since the available evidence on the effectiveness of ARISCAT and spirometry is scarce.” Page 22, lines 341-348. (clean file) 

7. In line 140 I suggest you specificate the model as mathematical model if it possible. 

Response: We appreciate the suggestion and now indicate that we used a “deterministic mathematical model” Page 9, line 160. (clean file) 

8. line 162 specificate the triangular distribution for estimating probabilities 

Response: As requested by the reviewer, we now indicate that the triangular distribution was used to estimate the probabilities of the post-operative complications and mortality. We agree that this change will improve the reader's understanding. Page 11, lines 189-190. (clean file) 

Reviewer #1

1-Consider expanding the description of the statistical techniques used. 

Response: We appreciate the suggestion to expand the description of the statistical techniques we used, and have included the following information in the Methods section:

“The ACER was calculated by dividing costs by effectiveness and the ICER was calculated as (Costs1-Costs2) / (Effectiveness1 - Effectiveness2). Post-operative complications in the spirometry group were calculated as follows for the ACER and ICER respectively:

POC ACER without spirometry =(Cost according to ARISCAT level without spirometry)/(Decrease in POC without spirometry) vs

POC ICER without spirometry=(Cost according to ARISCAT level without spirometry-with spirometry )/(Decrease in POC without spirometry- with spirometry)

These same formulas were applied to the survival results, for each group with and without spirometry, and for each ARISCAT risk level.” Pages 10-11, lines 179-187. (clean file) 

2-Consider expanding the data with criteria of complications and mortality. 

Response: We appreciate the suggestion, and have included more information about the criteria used for the post-operative complications and mortality. The following text was added to the Methods section: “The criteria data for post-operative pulmonary complications was defined according to the same criteria used by the PERISCOPE study (atelectasis, bronchospasm, pleural effusion, pneumonia, respiratory failure, pneumothorax, and pulmonary embolism) [10]. Other postoperative complications included, surgical (hypovolemic shock, sepsis, abdominal pain, fistula, bleeding, paralytic ileus, vascular injury, perforation); metabolic (glycemic dysregulation, hepatic or renal failure); cardiovascular (cardiogenic shock and acute myocardial infarction); neurological (acute vascular events); and vascular (deep vein thrombosis). Mortality was recorded if it was directly related to a complication from the surgery that was performed.” Pages 8-9, lines 140-148. (clean file)

---

## [Decision Letter · Decision Letter 1]

12 Jul 2022

Preoperative risk assessment and spirometry is a cost-effective strategy to reduce post-operative complications and mortality in Mexico

PONE-D-21-36131R1

Dear Dr. Yvonne N. Flores,

We’re pleased to inform you that your manuscript has been judged scientifically suitable for publication and will be formally accepted for publication once it meets all outstanding technical requirements.

Kind regards,

Carlos Alberto Zúniga-González, Ph.D

Academic Editor

PLOS ONE

Additional Editor Comments (optional):

Congratulations. I have checked that all observations reviewers have been incorporated.

Reviewers' comments:

Reviewer's Responses to Questions

**Comments to the Author**

1. If the authors have adequately addressed your comments raised in a previous round of review and you feel that this manuscript is now acceptable for publication, you may indicate that here to bypass the “Comments to the Author” section, enter your conflict of interest statement in the “Confidential to Editor” section, and submit your "Accept" recommendation.

Reviewer #1: All comments have been addressed

2. Is the manuscript technically sound, and do the data support the conclusions?

Reviewer #1: Yes

3. Has the statistical analysis been performed appropriately and rigorously? 

Reviewer #1: Yes

4. Have the authors made all data underlying the findings in their manuscript fully available?

Reviewer #1: Yes

5. Is the manuscript presented in an intelligible fashion and written in standard English?

Reviewer #1: Yes

6. Review Comments to the Author

Reviewer #1: the changes made to the document were reviewed, the observations were accepted, the document is ready for publication

7. PLOS authors have the option to publish the peer review history of their article (what does this mean?). If published, this will include your full peer review and any attached files.

Reviewer #1: No

---

## [Editor Report · Acceptance letter]

18 Jul 2022

PONE-D-21-36131R1 

Preoperative risk assessment and spirometry is a cost-effective strategy to reduce post-operative complications and mortality in Mexico 

Dear Dr. Flores:

I'm pleased to inform you that your manuscript has been deemed suitable for publication in PLOS ONE. Congratulations! Your manuscript is now with our production department. 

Kind regards, 

on behalf of

Dr. Prof. Carlos Alberto Zúniga-González 

Academic Editor

PLOS ONE